# Phase Coordinate Uncomputation in Quantum Recursive Fourier Sampling

**DOI:** 10.3390/e27060596

**Published:** 2025-06-02

**Authors:** Christoffer Hindlycke, Niklas Johansson, Jan-Åke Larsson

**Affiliations:** Department of Electrical Engineering, Linköping University, 581 83 Linköping, Sweden; niklas.johansson@liu.se (N.J.); jan-ake.larsson@liu.se (J.-Å.L.)

**Keywords:** quantum algorithms, phase kickback, quantum advantage, uncomputation, quantum oracles

## Abstract

Recursive Fourier Sampling (RFS) was one of the earliest problems to demonstrate a quantum advantage, and is known to lie outside the Merlin–Arthur complexity class. This work contains a new description of quantum algorithms in phase space terminology, demonstrating its use in RFS, and how and why this gives a better understanding of the quantum advantage in RFS. Most importantly, describing the computational process of quantum computation in phase space terminology gives a much better understanding of why uncomputation is necessary when solving RFS: the advantage is present only when phase coordinate garbage is uncomputed. This is the underlying reason for the limitations of the quantum advantage.

## 1. Introduction

Recursive Fourier Sampling (RFS) [1,2], originally introduced by Bernstein and Vazirani in 1993, was one of the earliest computational problems for which there existed a quantum algorithm apparently demonstrating an advantage over any classical algorithm. It is not known which complexity class RFS belongs to, but in the oracle paradigm it has been shown to lie outside Merlin–Arthur (**MA**) [3], which contains the Non-Deterministic Polynomial Time (**NP**) complexity class. Some further conjectures on which complexity class RFS may belong to are made in Ref. [4]. In spite of this, RFS has received scant attention, possibly in part owing to the implicit way in which it is formulated.

In this work, we provide a small generalization of RFS using updated notation, connect our generalization and previous formulations, and prove that uncomputation is necessary in a quantum solution to each RFS subproblem; this was already implied by existing complexity bounds. Our formulation does more since it enables a direct comparison with classical reversible computing where (computational coordinate) garbage is uncomputed to enable reversibility. In RFS, the uncomputation of phase coordinate garbage enables the quantum advantage. The mechanism of phase kickback [5,6,7] is an important quantum computational property that enables the quantum solution of RFS, in line with previous results [7] on, for instance, the Bernstein–Vazirani algorithm (which is equal to RFS for recursion depth 1). Here, we extend this picture to a full phase space description.

This is done by reformulating the oracles employed by RFS into the terminology of *conjugate pair oracles* that we define in this work; a conjugate pair oracle outputs a *conjugate pair* of two phase space coordinates, one in the computational basis and one in the phase basis. Describing and thereby understanding quantum computation in terms of phase space coordinates [8] has previously yielded a number of interesting results, such as the stabilizer formalism [9,10], several simple toy models, which nonetheless demonstrate a surprising number of quantum behaviors [7,11], and above all, quantum error-correcting codes [12,13,14]. This means of description thus allows for a rich representation while remaining accessible, in the sense that we may more easily understand what is happening during our computations. As RFS lies outside **MA**, our results lend credence to the conjecture [7] that oracle-paradigm bounds for the quantum advantage may be imposed by limited communication capacity rather than limited computational capacity: Any advantage seems to stem from the quantum oracle outputting (communicating) additional data compared to a classical oracle.

Below, we assume familiarity with basic linear algebra and quantum computation; for a comprehensive introduction, see [6]. We denote by $x_{k}$ a bitstring of some fixed length $n_{k}$, and use “·” to denote the dot product modulo 2. The rest of this work is organized as follows. In Section 2, we describe Fourier Sampling, how to perform it using classical and quantum machines and how it can be used to solve a certain decision problem; we also explain the quantum algorithm’s use of phase kickback. Here, we make use of standard notation. Next, in Section 3, we describe Recursive Fourier Sampling, and how to perform it using classical and quantum machines. The presentation is more streamlined than in earlier works: A comparison to earlier presentations can be found in Appendix A. Then, in Section 4, we go on to describe the contributions of the present work by introducing the notion of conjugate pair oracles, use this notion to show that uncomputing phase coordinate garbage is the enabling mechanism behind the quantum solution of RFS, and provide a proof of why uncomputation cannot be avoided. We conclude by summarizing our main findings, briefly discuss the complexity of RFS, and point out some future avenues of research.

## 2. Fourier Sampling and Phase Kickback

Given a function φ encoded into the coefficients of the quantum state $|\varphi \rangle =\sum_{x=0}^{N-1}\varphi \left(x\right)|x\rangle$, the Quantum Fourier Transform (QFT) gives(1)$$QFT|\varphi \rangle =\sum_{\chi =0}^{N-1}\hat{\varphi }\left(\chi \right)|\chi \rangle .$$
A computational basis measurement outcome *X* will then be a sample from the distribution $P\left(X=\chi \right)=|\hat{\varphi }{\left(\chi \right)|}^{2}$, known as Fourier Sampling, from the power spectrum of the function φ. Samples essentially give the inverse image of the power spectrum, e.g., frequencies absent from $\hat{\varphi }$ will occur with probability 0.

The bit-wise Quantum Fourier Transform mod 2 (the Hadamard transform) was used by Bernstein and Vazirani in 1993 [1] to solve the following problem: given an oracle for the Boolean function *f* promised to be linear, i.e.,(2)f(x)=s·x,
determine the unknown bitstring *s*. The linear choice of *f* is part of the original problem description. There is a version utilizing a non-linear (quadratic) *f* [15] but there exists no recursive variant of this problem.

A classical algorithm would need to call the corresponding classical reversible oracle(3)$$O_{f}\left(x,y\right)=\left(x,y+f(x)\right)$$
a total of *n* times (see Algorithm 1), since it can extract at most one bit of information from each call, available in the target bit *y*. Querying *f* with the argument $x=1_{j}$ where $1_{j}$ is the bitstring with only the *j*th bit set, the function output is ${\left(s\right)}_{j}$, the *j*th bit of the bitstring *s*.
**Algorithm 1** Classical Fourier Sampling
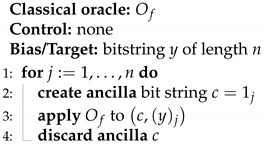


The promise (Equation 2) tells us that if the initial bias y=0, the output target *y* will have the value *s*, since $f\left(1_{j}\right)=s\cdot 1_{j}={\left(s\right)}_{j}$.

An appropriate function for Fourier Sampling is ${\left(-1\right)}^{f(x)}$ because the Hadamard transform of ${\left(-1\right)}^{f(x)}=1-2f\left(x\right)$ is $1-2\hat{f}\left(\chi \right)=\delta_{\chi s}$, the Kronecker delta function, which equals 1 if χ=s and 0 otherwise. If one can generate a quantum state containing ${\left(-1\right)}^{f(x)}$ in its coefficients, Fourier Sampling will give *s* (with probability 1). Quantum algorithms provide a way to accomplish this through a procedure known as phase kickback [5,7].

Phase kickback uses a quantum function oracle $U_{f}$ that preserves phase information,(4)$$U_{f}\sum_{x,y}c_{x,y}|x,y\rangle =\sum_{x,y}c_{x,y}|x,y+f(x)\rangle ,$$
and elements of the phase basis in control and target registers, to move the function output from the target computational basis state in the right-hand side above, to the phase of the coefficients in the controlling register. Applying $U_{f}$ to the phase basis elements gives(5)$$\begin{array}{cc} U_{f}H^{⊗n+1}|\chi ,\upsilon \rangle & =U_{f}\sum_{x,y}{\left(-1\right)}^{\chi \cdot x}|x\rangle {\left(-1\right)}^{\upsilon y}|y\rangle =\sum_{x,y}{\left(-1\right)}^{\chi \cdot x}|x\rangle {\left(-1\right)}^{\upsilon y}|y+f(x)\rangle \\ \; & =\sum_{x,y}{\left(-1\right)}^{\chi \cdot x}|x\rangle {\left(-1\right)}^{\upsilon (y-f(x))}|y\rangle =\sum_{x,y}{\left(-1\right)}^{\chi \cdot x-\upsilon f(x)}|x\rangle {\left(-1\right)}^{\upsilon y}|y\rangle . \end{array}$$
For Quantum Fourier Sampling, we use the Hadamard transform and bitwise addition modulo 2, and the Fourier variables are bitstrings x,χ of length *n* and bits y,υ. Then, the Fourier Sampling promise of Equation (Equation 2) gives(6)$$\begin{array}{c} H^{⊗n+1}U_{f}H^{⊗n+1}|\chi ,\upsilon \rangle =H^{⊗n+1}\sum_{x,y}{\left(-1\right)}^{(\chi +\upsilon s)\cdot x}|x\rangle {\left(-1\right)}^{\upsilon y}|y\rangle =|\chi +\upsilon s,\upsilon \rangle . \end{array}$$

Bernstein and Vazirani [1] write this as an algorithm that solves the Fourier Sampling problem in one call to the quantum oracle, see Algorithm 2.
**Algorithm 2** Quantum Fourier Sampling**Quantum oracle:** 
$U_{f}$**Control:** none**Bias/Target:** qubit string $|\psi_{x}\rangle$ of length *n*1:**create ancilla** qubit $|\psi_{y}\rangle :=|1\rangle$2:**apply** $H^{⊗n+1}$ to $|\psi_{x},\psi_{y}\rangle$3:**apply** $U_{f}$ to $|\psi_{x},\psi_{y}\rangle$4:**apply** $H^{⊗n+1}$ to $|\psi_{x},\psi_{y}\rangle$5:**discard ancilla** 
$|\psi_{y}\rangle$

The promise (Equation 6) tells us that if the initial bias $|\psi_{x}\rangle =|\chi \rangle =|0\rangle$, measurement of the output $|\psi_{x}\rangle$ will give the outcome *s* (with probability 1), and also that the ancilla is fixed to a known value in step 5 so can be safely discarded.

The problem as stated so far is a sampling problem: sample from a specific probability distribution (output *s* with probability 1). In what follows, it will be useful to re-state the problem as a decision problem, that determines if a given bitstring $x_{1}$ belongs to the language $L_{f_{1}}$. In the decision problem version of Fourier Sampling, we are given a Boolean function of two arguments, promised to obey(7)$$f_{2}\left(x_{1},x_{2}\right)=s_{1}\left(x_{1}\right)\cdot x_{2},$$
and another Boolean function $g_{1}\left(x_{1},s_{1}\left(x_{1}\right)\right)$ that gives the answer to the problem upon input of $s_{1}\left(x_{1}\right)$ (this latter function can be referred to as $f_{1}\left(x_{1}\right)$). Given oracle access to $f_{2}$ and $g_{1}$, we need to use classical or Quantum Fourier Sampling on the last argument of $f_{2}$ to obtain $s_{1}\left(x_{1}\right)$ so that we can use $g_{1}$ to determine if $x_{1}$ belongs to $L_{f_{1}}$ or not, to solve the decision problem.

Note that if $g_{1}$ is linear in the second argument and independent of the first, then there exists an $x^{*}$ so that $g_{1}\left(x_{1},x_{2}\right)=x^{*}\cdot x_{2}$, and $f_{1}\left(x_{1}\right)=g_{1}\left(x_{1},s_{1}\left(x_{1}\right)\right)=x^{*}\cdot s_{1}\left(x_{1}\right)=f_{2}\left(x_{1},x^{*}\right)$. Such a decision problem can be solved in a single call to $f_{2}$; thus problems with a linear $g_{1}$ (sometimes known as a parity function) are less interesting to study [1].

## 3. Recursive Fourier Sampling

There are, to the best of our knowledge, three formulations of RFS. The original was introduced by Bernstein and Vazirani in 1993 [1], further expanded on by the same authors in 1997 [2], and presented in a slightly different manner by Vazirani in 2002 [3]. In 2003, Aaronson [16] re-formulated RFS somewhat, in part by removing the bitstring size limitations. Finally, in 2008 Johnson [4] essentially combined elements of the two versions in another formulation. We shall make use of the definitions of Ref. [3], but remove the bitstring size limitations in keeping with [16], and add subscripts to function sequences for indexing the recursion level. A comparison of the notation for the different versions can be found in Appendix A.

RFS is a decision problem: it uses a so-called Fourier Sampling tree $\left\{f_{k}:1\leq k\leq l+1\right\}$ to decide whether a bitstring $x_{1}$ belongs to a language $L_{f_{1}}$ or not. A Fourier Sampling tree is a Boolean function sequence such that for k>1(8)$$f_{k}\left(x_{1},\ldots ,x_{k}\right)=s_{k-1}\left(x_{1},\ldots ,x_{k-1}\right)\cdot x_{k}.$$
We say that the sequence $f_{k}$ is *derived from* the sequence $g_{k}$ for 1≤k≤l (and $g_{k}$
*specifies* the Fourier Sampling tree $f_{k}$) if(9)$$\begin{array}{c} f_{k-1}(x_{1},\ldots ,x_{k-1})=g_{k-1}\left(x_{1},\ldots ,x_{k-1},s_{k-1}\left(x_{1},\ldots ,x_{k-1}\right)\right). \end{array}$$
Now, given oracles for $g_{k}$, 1≤k≤l, and $f_{l+1}$, the challenge is to determine $f_{1}\left(x_{1}\right)$, which in turn, tells us if $x_{1}$ is in the language $L_{f_{1}}$ or not. Note that oracle access to $f_{l+1}$, though not explicitly stated in [3], is necessary to perform Quantum Fourier Sampling as described there.

A classical algorithm would need to solve this recursively, at each step using the corresponding classical reversible oracle(10)$$O_{f_{k}}\left(x_{1},\ldots ,x_{k},y\right)=\left(x_{1},\ldots ,x_{k},y+f\left(x\right)\right)$$
a total of $n_{k}$ times, since it can only extract (at most) one bit of information from each call, available in the target bit *y*. We here rewrite the explicit classical algorithm by McKague [17] in our notation, see Algorithm 3.
**Algorithm 3** Classical Recursive Fourier Sampling
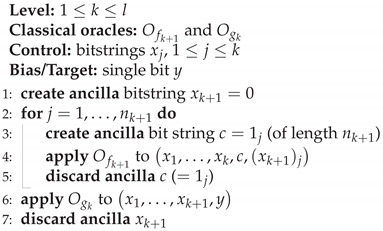


From the promise (Equation 2), it follows that oracle access to $f_{k+1}$ gives us $f\left(x_{1},\ldots ,x_{k},1_{j}\right)={\left(s_{k}\left(x_{1},\ldots ,x_{k}\right)\right)}_{j}$, so that $x_{k+1}=s_{k}\left(x_{1},\ldots ,x_{k}\right)$ after the recursion for-loop. It then follows from the promise (Equation 9) that oracle access to $g_{k}$ with this value of $x_{k+1}$ gives oracle access to $f_{k}$. That this holds for all *k* follows by induction, in particular, we obtain oracle access to $f_{1}$ so that we can calculate $f_{1}\left(x_{1}\right)$ [17].

This algorithm needs $n_{2}n_{3}\ldots n_{l+1}$ calls to $f_{l+1}$ and $n_{2}n_{3}\ldots n_{k}$ calls to $g_{k}$, 1<k≤l, so the algorithm has the complexity $O\left(\prod_{k=2}^{l+1}n_{k}\right)$, which in the case of equal lengths $n_{k}=n$ is $O(n^{l})$ [16]; the corresponding lower bound is then $\Omega (n^{l})$ [4], meaning we have a strict bound $\Theta (n^{l})$. Calculating an exact bound for a problem given string lengths $n_{1},\ldots ,n_{k}$ is easily done via the formula above.

A quantum oracle would, also here, be assumed to preserve phase information(11)$$\begin{array}{cc} \; & U_{f_{k}}\sum_{x_{1},\ldots ,x_{k},y}c_{x_{1},\ldots ,x_{k},y}|x_{1},\ldots ,x_{k},y\rangle =\sum_{x_{1},\ldots ,x_{k},y}c_{x_{1},\ldots ,x_{k},y}|x_{1},\ldots ,x_{k},y+f\left(x_{1},\ldots ,x_{k}\right)\rangle . \end{array}$$
Let $H_{j}$ denote the unitary that applies Hadamards on the qubits of $|x_{j}\rangle$. Then, the Fourier Sampling promise of Equation (Equation 8), with parameters $x_{1},\ldots ,x_{k-1}$, gives(12)$$\begin{array}{c} \left(H_{k}⊗H\right)U_{f_{k}}\left(H_{k}⊗H\right)|x_{1},\ldots ,x_{k-1},\chi ,\upsilon \rangle =|x_{1},\ldots ,x_{k-1},\chi +\upsilon s_{k-1}\left(x_{1},\ldots ,x_{k-1}\right),\upsilon \rangle . \end{array}$$
The assumption (Equation 11) implies phase is preserved for a superposition of such input states. A quantum algorithm can now use such a quantum oracle and a different recursion [17], see Figure 1 and Algorithm 4.
**Algorithm 4** Quantum Recursive Fourier Sampling**Level:** 
1≤k≤l**Quantum oracles:** $U_{f_{k+1}}$ and $U_{g_{k}}$**Controls:** qubit strings $|\psi_{x_{j}}\rangle$, 1≤j≤k**Bias/Target:** single qubit $|\psi_{y}\rangle$1:**create ancilla** qubit string $|\psi_{x_{k+1}}\rangle :=|0\rangle$ and qubit $|\psi_{y^{\prime }}\rangle :=|1\rangle$2:**apply** $H^{⊗n_{k+1}+1}$ to $|\psi_{x_{k+1}},\psi_{y^{\prime }}\rangle$3:**apply** $U_{f_{k+1}}$ to $|\psi_{x_{1}},\ldots ,\psi_{x_{k+1}},\psi_{y^{\prime }}\rangle$4:**apply** $H^{⊗n_{k+1}}$ to $|\psi_{x_{k+1}}\rangle$5:**apply** $U_{g_{k}}$ to $|\psi_{x_{1}},\ldots ,\psi_{x_{k+1}},\psi_{y}\rangle$6:**apply** $H^{⊗n_{k+1}}$ to $|\psi_{x_{k+1}}\rangle$7:**apply** $U_{f_{k+1}}$ to $|\psi_{x_{1}},\ldots ,\psi_{x_{k+1}},\psi_{y^{\prime }}\rangle$8:**apply** $H^{⊗n_{k+1}+1}$ to $|\psi_{x_{k+1}},\psi_{y^{\prime }}\rangle$9:**discard ancillas** 
$|\psi_{x_{k+1}},\psi_{y^{\prime }}\rangle$

With the ancillas $|\psi_{x_{k+1}},\psi_{y^{\prime }}\rangle =|\chi ,\upsilon \rangle =|0,1\rangle$, it follows from Equation (Equation 12) that $|\psi_{x_{k+1}}\rangle =|s_{k}\left(x_{1},\ldots ,x_{k}\right)\rangle$ after step 4. It then follows from the promise (Equation 9) that oracle access to $g_{k}$ through $U_{g_{k}}$ with this state $|\psi_{x_{k+1}}\rangle$ gives oracle access to $f_{k}$, creating $U_{f_{k}}$. Then steps 6–8 reset $|\psi_{x_{k+1}}\rangle =|0\rangle$ so that it can be safely discarded, usually motivated by the need to enable interference [2,16]; the actual underlying reason is to ensure that phase preservation of $U_{g_{k}}$ gives phase preservation of $U_{f_{k}}$, we will return to this point below. That we have oracle access to $f_{k}$ for all *k* follows by induction, in particular, we obtain oracle access to $f_{1}$ so that we can calculate $f_{1}\left(x_{1}\right)$ [17]. c This algorithm needs $2^{l}$ calls to $U_{f_{l+1}}$, so the algorithm has the complexity $O(2^{l})$, and the corresponding lower bound is $\Omega (2^{l})$ [4,16], meaning we have a strict bound $\Theta (2^{l})$.

## 4. Conjugate Pair Oracles and Phase Coordinate Uncomputation

We are now almost in a position to define and use conjugate pair oracles. This will yield an alternative description of RFS, in turn, enabling a proof that uncomputation is necessary in RFS, which (together with the conjugate pair formalism) is the main contribution of this work. As a conjugate pair oracle acts on a conjugate pair, we begin by defining the latter, as a representation based on phase space coordinates [8] and this idea allows for our results.

The Hadamard arrangement of the presented quantum algorithms is usually only seen as a convenient way to move *f* from the target register into the phase of the coefficients of the control register, to enable Fourier Sampling from that register [1]. The $|\psi_{y}\rangle =|\upsilon \rangle$ bias input is converted into a negative phase ${\left(-1\right)}^{\upsilon }$, which then enables the “phase kickback” of the function output into the phase of the control system, to enable the $|\psi_{y}\rangle =|s\rangle$ output if υ=1. However, this can be viewed in a different manner. The Fourier coordinate is a canonically conjugate phase space coordinate, often referred to as “the phase basis”.

In textbooks on quantum mechanics, one typically first encounters the canonically conjugate pair [x,p] of position *x* and momentum *p*. The brackets are here meant to indicate “the canonically conjugate pair” of physical quantities, not the commutator between two Hermitian operators as is commonplace in quantum mechanics. In quantum computing, the appropriate conjugate pair is computational basis and phase basis, the pair X=[x,χ] of computational basis bitstring *x* and phase bitstring χ, of equal length. It is this which we take as our definition of a conjugate pair, two bitstrings of equal length; this is then the length of the conjugate pair.

In quantum mechanics, if the state of the studied system is an element of the computational basis, then *x* is well-defined but χ is not; if the state is an element of the phase basis, then χ is well-defined but *x* is not. This is a consequence of the uncertainty relation [7]. Even partial knowledge of χ forces the value of *x* to be partially unknown, or really, partially unknowable, and vice versa. Therefore, we cannot access both entries of the conjugate pair X=[x,χ], we must choose one of them. In other words: for a conjugate pair (when used in quantum computation) we may have that the computational basis bitstring is well-defined, or that the phase bitstring is well-defined, or that neither is well-defined (but never that both are well-defined).

Creating a computational basis state can now be seen as writing a well-defined value *x* into the computational basis part of the conjugate pair X=[x,χ], and measuring in the computational basis can be seen as reading off the *x* value. Likewise, creating a phase basis state can now be seen as writing a well-defined value χ into the phase basis part of the conjugate pair X=[x,χ], and measuring in the phase basis can be seen as reading off the χ value.

The Hadamard operation or bitwise Fourier transform mod 2 moves a potential well-defined value from the computational basis part to the phase basis part of the state and vice versa; an appropriate notation for a system where X has length *n* would be(13)$$H^{n}X=H^{n}\left[x,\chi \right]=\left[\chi ,x\right].$$

Re-writing Quantum Fourier Sampling from this point of view, the first two steps of the algorithm write information into the phase basis, and the final two steps read information from the phase basis [7]. The promise associated with a quantum oracle, that phase is preserved among the components of a superposition, now implies a simpler promise: that the oracle can calculate not one but two kinds of output. The standard function map(14)x↦f(x),
is still available, with the standard control and bias/target registers. However, a second function map is also available, the phase kickback map that we will denote(15)$$\upsilon \mapsto \overset{⌵}{f}(\upsilon ),$$
from the phase coordinate of the target/bias register to the phase coordinate of the control. The behavior of these two function maps can now be collected into a joint description: a *conjugate pair oracle* of the form in the following theorem.

**Theorem 1.** 
*Under the promise *(Equation 4)* of phase preservation of the quantum function oracle, we can write down the pair of maps in the computational basis and phase basis, respectively, as the conjugate pair oracle*

(16)
$$F\left([x,\chi ],[y,\upsilon ]\right)=\left(\left[x,\chi +\overset{⌵}{f}(\upsilon )\right],\left[y+f(x),\upsilon \right]\right).$$

*where $\overset{⌵}{f}(\upsilon )$ is random with*

(17)
$$\overset{⌵}{f}\left(\upsilon \right)=\upsilon T,\;p\left(T=t\right)={\left|\delta_{t0}-2\hat{f}\left(t\right)\right|}^{2}$$

*so that $\overset{⌵}{f}(\upsilon )$ is a Fourier sample of f when υ=1, thus the notation $\overset{⌵}{f}(\upsilon )$ with a checkmark (the Kronecker delta function $\delta_{t0}$ equals 1 when t equals the all-zero vector, and the Fourier transform used is the Hadamard transform).*


**Proof.** The computational coordinate map immediately follows from the function map of the quantum oracle. The phase coordinate map is obtained from phase preservation (Equation 4) and the identity ${\left(-1\right)}^{w}=1-2w$ (skipping the steps already present in Equation (Equation 6)), through(18)$$\begin{array}{cc} H^{⊗n+1}U_{f}H^{⊗n+1}|\chi ,\upsilon \rangle & =H^{⊗n+1}\sum_{x,y}{\left(-1\right)}^{\chi \cdot x+\upsilon f(x)}|x\rangle {\left(-1\right)}^{\upsilon y}|y\rangle \\ & =H^{⊗n+1}\sum_{x,y}{\left(-1\right)}^{\chi \cdot x}\left(1-2\upsilon f(x)\right)|x\rangle {\left(-1\right)}^{\upsilon y}|y\rangle \\ & =\sum_{t,x}{\left(-1\right)}^{(\chi +t)\cdot x}\left(1-2\upsilon f(x)\right)|x,\upsilon \rangle \\ \; & =\sum_{t}\left(\delta_{t0}-2\upsilon \hat{f}\left(t\right)\right)|\chi +t,\upsilon \rangle \end{array}$$
Therefore, the phase bit υ of the target system is unchanged, while the phase bitstring χ of the control is unchanged if υ=0, otherwise shifted with a random Fourier sample distributed as indicated in Equation (Equation 17).    □

The above should be read as follows: if the input computational basis bitstrings x,y are well-defined, the output computational basis bitstrings x,y+f(x) are well-defined so that f(x) can be deduced. Furthermore, if instead, the input phase basis bitstrings χ,υ are well-defined, the output phase basis bitstrings $\chi +\overset{⌵}{f}(\upsilon ),\upsilon$ are well-defined so that $\overset{⌵}{f}(\upsilon )$ can be deduced. By our definition of a conjugate pair, it is clear that only one part of the conjugate pair oracle F can be queried at any one time.

In Fourier sampling, the distribution of $\overset{⌵}{f}$
is especially simple. There is an additional promise on the structure of *f* in Equation (Equation 2), which in conjunction with phase preservation, using Theorem 1, gives(19)$$\overset{⌵}{f}(\upsilon )=\upsilon s$$
with probability 1. Here, we must stress that the simplicity of this is a direct consequence of phase preservation (Theorem 1) and the additional promise of Equation (Equation 2). We can now solve the Fourier Sampling problem by just accessing this function in the phase information, this gives us Algorithm 5.
**Algorithm 5** Conjugate Pair Oracle Fourier Sampling**Conjugate pair oracle:** 
F**Control:** none**Bias/Target:** conjugate pair X1:**create ancilla** Y:=[y,υ] of length 1 with υ:=12:**apply** F to (X,Y)3:**discard ancilla** 
Y

**Theorem 2.** 
*For a target conjugate pair X=[x,χ] where the phase bitstring χ=0, Algorithm 5 solves Fourier Sampling. The solution is available after the algorithm as the phase bitstring χ.*


**Proof.** If the input bias conjugate pair X=[x,χ] has well-defined phase χ=0, it follows from Equation (Equation 19) that the output conjugate pair X=[x,χ] has well-defined phase χ=s with probability 1.    □

The output χ is a bitstring of length *n*, so querying the phase part of the oracle can provide *n* bits of information in one use. This is obviously impossible for a classical oracle that returns a Boolean output.

Quantum Recursive Fourier Sampling can also be re-written from this point of view, adding some complication because of the *k* arguments $x_{j}$; each phase coordinate $\chi_{j}$ will now be shifted with a separate ${\overset{⌵}{f}}_{k,j}$. Theorem 1 generalizes in the following way.

**Theorem 3.** 
*Under the promise *(Equation 11)* of phase preservation of the quantum function oracle, we can write down the pair of maps in the computational basis and phase basis, respectively, as the conjugate pair oracle*

(20)
$$\begin{array}{cc} \; & F_{k}\left(\left[x_{1},\chi_{1}\right],\ldots ,\left[x_{k},\chi_{k}\right],\left[y,\upsilon \right]\right)\\ \; & =(\left[x_{1},\chi_{1}+{\overset{⌵}{f}}_{k,1}\left(\upsilon ,x_{2},\ldots ,x_{k}\right)\right],\left[x_{2},\chi_{2}+{\overset{⌵}{f}}_{k,2}\left(x_{1},\upsilon ,x_{3}\ldots ,x_{k}\right)\right],\ldots ,\\ \; & \;\left[x_{k},\chi_{k}+{\overset{⌵}{f}}_{k,k}\left(x_{1},\ldots ,x_{k-1},\upsilon \right)\right],\left[y+f_{k}\left(x_{1},\ldots ,x_{k}\right),\upsilon \right]). \end{array}$$

*where ${\overset{⌵}{f}}_{k,j}\left(x_{1},\ldots ,x_{j-1},\upsilon ,x_{j+1},\ldots ,x_{k}\right)$ is random with*

(21)
$$\begin{array}{cc} {\overset{⌵}{f}}_{k,j}\left(x_{1},\ldots ,x_{j-1},\upsilon ,x_{j+1},\ldots ,x_{k}\right) & =\upsilon T_{j}\left(x_{1},\ldots ,x_{j-1},x_{j+1},\ldots ,x_{k}\right),\\ p\left(T_{j}\left(x_{1},\ldots ,x_{j-1},x_{j+1},\ldots ,x_{k}\right)=t\right) & ={\left|\delta_{t0}-2{\hat{f}}_{k,j}\left(x_{1},\ldots ,x_{j-1},t,x_{j+1},\ldots ,x_{k}\right)\right|}^{2}. \end{array}$$

*Here, ${\hat{f}}_{k,j}$ is the Fourier transform on the j:th argument of $f_{k}$, so that ${\overset{⌵}{f}}_{k,j}$ is a Fourier sample on the j:th argument of f when υ=1.*


**Proof.** For each *j*, use Theorem 1 on the conjugate pair oracle function $F_{k}(X_{1},\ldots ,X_{j-1},X,$$X_{j+1},\ldots ,X_{k},Y)$. Theorem 1 applies because this converts the phase preservation of Equation (Equation 11) into that of Equation (Equation 4).    □

In RFS, we also have an additional promise on the structure of $f_{k}$ in Equation (Equation 8), but only for the last argument, and in conjunction with phase preservation, using Theorem 3, we obtain(22)$${\overset{⌵}{f}}_{k,k}\left(x_{1},\ldots ,x_{k-1},\upsilon \right)=\upsilon s_{k-1}\left(x_{1},\ldots ,x_{k-1}\right)$$
with probability 1. Again, we must stress that the simplicity of this is a direct consequence of phase preservation (Theorem 3) and the additional promise of Equation (Equation 8).

The other ${\overset{⌵}{f}}_{k,j}$ are random and depend on $x_{i}$, i≠j, but we have no specific promised distribution when j<k; this is simply not part of the problem description. Even so, we can solve Recursive Fourier Sampling by just accessing this function in the phase information of the *k*:th control. Note that a random value ${\overset{⌵}{f}}_{k,j}$, j<k is added to each $\chi_{j}$, j<k during this process, which needs to be removed to preserve the value of $\chi_{j}$ when calculating the next level. This gives us Algorithm 6.
**Algorithm 6** Conjugate Pair Oracle Recursive Fourier Sampling**Level:** 
1≤k≤l**Conjugate pair oracles:** $F_{k+1}$ and $G_{k}$**Controls:** conjugate pairs $X_{j}$, 1≤j≤k**Bias/Target:** conjugate pair Y1:**create ancillas** $X_{k+1}=\left[x_{k+1},\chi_{k+1}\right]$ with $\chi_{k+1}:=0$ and $Y^{\prime }=\left[y^{\prime },\upsilon^{\prime }\right]$ with $\upsilon^{\prime }:=1$2:**apply** $F_{k+1}$ to $\left(X_{1},\ldots ,X_{k+1},Y^{\prime }\right)$3:**apply** $H^{n_{k+1}}$ to $X_{k+1}$4:**apply** $G_{k}$ to $\left(X_{1},\ldots ,X_{k+1},Y\right)$5:**apply** $H^{n_{k+1}}$ to $X_{k+1}$6:**apply** $F_{k+1}$ to $\left(X_{1},\ldots ,X_{k+1},Y^{\prime }\right)$7:**discard ancillas** 
$\left(X_{k+1},Y^{\prime }\right)$

**Theorem 4.** 
*For a target conjugate pair Y=[y,υ] with phase bitstring υ=1, Algorithm 6 starting at level *1* solves Recursive Fourier Sampling. The solution is available after the algorithm as the phase bitstring $\chi_{1}$.*


**Proof.** At level *k*, the proof has two parts: that oracle access to $F_{k+1}$ and $G_{k}$ give oracle access to $F_{k}$; and that the algorithm leaves the phase kick-back of $F_{k}$ in the arguments untouched. First, the ancillas $\left(X_{k+1},Y^{\prime }\right)$ have well-defined phases (0,1), so it follows from Equation (Equation 22) that $X_{k+1}$ has well-defined phase $s_{k}\left(x_{1},\ldots ,x_{k}\right)$ after applying $F_{k+1}$ the first time in step 2. The Hadamard of step 3 moves this into a well-defined computational basis value. It then follows from the promise (Equation 9) that oracle access to $G_{k}$ with this conjugate pair $X_{k+1}$ gives oracle access to $F_{k}$, including the phase kickback map. Second, the transformation in step 6 is the exact inverse of the transformation in step 2, so the second addition will remove precisely the values added in step 2, so that $\chi_{j}$, j<k+1 only contains the phase kickback of $F_{k}$.That we have oracle access to $F_{k}$ for all *k* follows by induction, in particular, we obtain oracle access to $F_{1}$ so that we can deduce $f_{1}\left(x_{1}\right)$ from $F_{1}\left(\left[x_{1},\chi_{1}\right],\left[y,\upsilon \right]\right)=\left(\left[x_{1},\chi_{1}+\overset{⌵}{f_{1}}\left(\upsilon \right)\right],\left[y+f_{1}\left(x_{1}\right),\upsilon \right]\right)$ by using the well-defined input $x_{1}$ and bias y=0. □

The crucial point here is that in this formulation of the problem and solution algorithm, although technically equivalent to the quantum formulation, steps 5 and 6 are no longer motivated by the somewhat unclear need to “uncompute ‘garbage’ [in the computational coordinate of the ancillary systems] left over by the first call, and thereby enable [proper] interference” [2,16]. Here, the motivation is much clearer: We need to uncompute the shift of *the phase coordinate of the controlling systems* with indices j<k+1, i.e., uncompute *phase coordinate garbage*, see Figure 2. This enables a direct comparison with classical reversible computing where (computational coordinate) garbage is uncomputed to enable reversibility. In RFS, the uncomputation of phase coordinate garbage enables the quantum advantage.

**Corollary 1.** 
*When using Algorithm 6 to solve Recursive Fourier Sampling, the uncomputation in step 6 is necessary to uncompute phase coordinate garbage added to $X_{j}$, j<k+1 in step 2.*


**Proof.** Step 2 of Algorithm 6 adds random values to the phase bitstrings of $X_{j}$, j<k+1 according to Equation (Equation 20). Step 6 is necessary to subtract values that are identical to the added values. □

The reason we need to perform uncomputation, to invert the addition of these additional phases, is that we are not guaranteed anything about the nature of ${\overset{⌵}{f}}_{k,j}$, j<k+1, by the problem formulation. Adding such a guarantee in the problem statement could give a further quantum advantage, although that would require maintaining the classical complexity of the problem under such an addition. One possible addition is promising linear $g_{k}$ at every level, but as mentioned earlier, that would reduce the classical complexity considerably [1].

## 5. Conclusions

In this work, the main contribution is the notion of conjugate pair oracles, which enables a reformulation of RFS, in turn, providing a stronger argument for why uncomputation is necessary than previously available. We also provide a small generalization of Recursive Fourier Sampling: our generalized RFS contains both the original formulation by Bernstein and Vazirani [3] and the one by Aaronson [16] as special cases.

Uncomputation is needed because the function oracle (quantum or conjugate pair, both with phase kickback) adds random phase shifts, i.e., computational garbage, to the controlling registers. Furthermore, we are not guaranteed anything about the value or distribution of these phase shifts by the problem formulation; the only guarantee we have concerns the phase shift of the last argument. Adding further guarantees in the problem statement of the behavior of the involved functions could give a further quantum advantage, although that would require maintaining the classical complexity of the problem under such an addition; we conjecture that such a modification is possible.

It should be noted here that the complexity bound on solving RFS, i.e., the exact bound $\Theta (2^{l})$ mentioned at the end of Section 3 [4,16], already indicates that uncomputation is needed. However, phase kickback in the conjugate pair oracle paradigm presented here arguably gives a better understanding of the necessity of uncomputation of phase coordinate garbage within the Quantum RFS algorithm.

It has previously been established [7] that at least some problems in **NP** relative to an oracle (such as Simon’s algorithm [2]) use phase kickback as their driving mechanism. We now have that RFS, which in a very real sense is “more difficult” than Simon’s algorithm given that RFS lies outside **MA** relative to an oracle [3], relies on this very same property of quantum systems as the enabling computational resource. That is, in the oracle paradigm, it is not so much a matter of computational power enabling an advantage, but rather one of communication: accessibility of the relevant information outside the oracle. As long as the correct phase information is accessible, we can (apparently) solve extremely hard problems more efficiently than otherwise. This strengthens the argument [7] that phase kickback is an important property of quantum computational systems. Clearly, phase kickback is critical in enabling a quantum advantage in RFS.

This formalism gives a new understanding of the behavior of quantum algorithms such as Grover’s algorithm [18] or Shor’s algorithm [19], in that it allows the explicit tracking of the phase information in a quantum circuit [7,10]. It also points out a possible path towards proving a practically meaningful distinction between classical and quantum computation. Perhaps the difficulty in classically simulating quantum computation stems entirely from the difficulty in maintaining the correct phase map in a quantum computation. Investigating this venue more thoroughly may well lead to the unconditional separation theorem that the quantum information society is still striving for to this day.

## Figures and Tables

**Figure 1 entropy-27-00596-f001:**
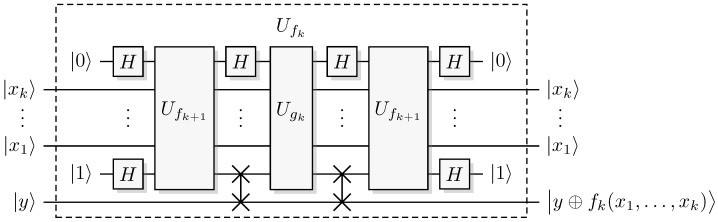
Quantum Recursive Fourier Sampling. The focus of the uncomputation is to reset the ancillary systems $|x_{k+1}\rangle$ to |0〉 and $|y^{\prime }\rangle$ to |1〉. The focus changes and becomes much clearer in the conjugate pair oracle paradigm in Section 4.

**Figure 2 entropy-27-00596-f002:**
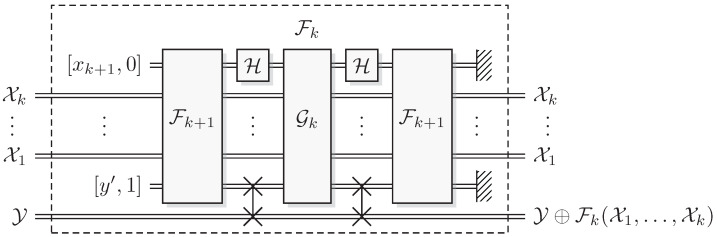
Conjugate Pair Recursive Fourier Sampling. The focus of the uncomputation is to undo the shift of the phase coordinate of $X_{1}$, *…*, $X_{k}$ from the $F_{k+1}$ oracle.

## Data Availability

No new data were created or analyzed in this study. Data sharing is not applicable to this article.

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
