# Peer review of "Phase Coordinate Uncomputation in Quantum Recursive Fourier Sampling"

_entropy, 2025, doi:10.3390/e27060596_

Round 1

Reviewer 1 Report

Comments and Suggestions for Authors

The manuscript proposed a quantum algorithm in phase kickback and its use in RFS. We have the following comments:

(1) “[]” denotes commutative operation in quantum computation, and denotes in manuscript as the canonically conjugate pair. Is there a better way to express the latter?

(2) Algorithms 4 and 6 should give the result in each step, for example, in Algorithm 6, apply H^{n_k+1} to X_{k+1}, then write the result at the end.

(3) The manuscript should give a experiment to test the proposed method. Python+Qiskit is suggested to use. We think 8-qubits can be used in a usual computer.

(4) In short, the manuscript has a strong reasoning process of quantum computing. We hope that the authors can also give a quantum circuit in Sec.4 similar to Fig. 1.

Reviewer 2 Report

Comments and Suggestions for Authors

I have read ``Phase Coordinate Uncomputation in Quantum Recursive Fourier Sampling'' by Hindlycke, Johansson and Larsson, manuscript 3614588, with interest. This paper concerns the computational problem Recursive Fourier Sampling, which is an oracle problem with no efficient classical algorithm (in the query complexity sense), but with a tractable algorithm on a quantum computer: the famous Bernstein-Vazirani algorithm. This problem has been extensively studied before, but the authors put forward a particular version of the algorithm to make two new arguments: first, that the algorithm can be understood in terms of the idea of a ``conjugate pair of oracles,'' and second, that this understanding explains why it is necessary to uncompute part of the circuit to realize quantum advantage.

The idea of a conjugate pair of oracles is that an oracle in the standard reversible form---which classically computes some black-box function, possibly with an additional promise---in the quantum case can be thought of as computing either of *two* possible functions. The first is the same function as in the classical case, which is done by submitting a system in a computational basis state to the oracle; the second is the ``conjugate'' function, that can be evaluated by submitting a state in a transformed basis. The conjugate function shows up in the phase of the components of the state, and can be measured by applying the inverse transform after the oracle call. This transform could be a quantum Fourier transform; in the case of the Bernstein-Vazirani algorithm (as well as others, like the Deutsch-Josza and Simon algorithms) it is a Hadamard transform.

While this idea has been used in a variety of quantum algorithms (as mentioned above), I have never seen it expressed in terms of a conjugate function as done in this paper. To the best of my knowledge this is an original contribution. (I do not, however, primarily work on quantum complexity theory, so my knowledge could be lacking.) It's an interesting way to think about how these oracle-based algorithms work, and could possibly be fruitful in thinking about new algorithms.

As for the necessity of uncomputation, I have not seen this specific argument before, but I think the necessity of uncomputation in many quantum algorithms is pretty well accepted. It's not entirely clear to me what has been added in this new argument over what was known before.

In general, I thought this paper was interesting and pretty well-written. I do not mostly work on quantum complexity theory, so I would feel more comfortable if it could also be assessed by some who works mainly in that field. Based on what I know this paper seems publishable. However, I would strongly urge the authors to clarify what they have added in their argument about the necessity of uncomputation. What exactly have they shown that was not known before?

In addition to that, I did have a couple of other minor comments. First, in a few places the authors say that ``Recursive Fourier Sampling has been proven to be outside MA'' (Merlin-Arthur). This proof is *relative to an oracle*. (It would be pretty exciting if it had been proven for a non-oracle problem!) I'm sure the authors thought that was obvious, since RFS is an oracle problem, but I think it is important to be clear.

I also noticed an odd terminology on p2: the sections of a paper are called "sections," not "chapters."

Reviewer 3 Report

Comments and Suggestions for Authors

The manuscript “Phase coordinate uncomputation in quantum recursive Fourier sampling” by C. Hindlycke, N. Johansson, and J.-A. Larsson provides a small generalization of Recursive Fourier Sampling (RFS) with the formulations of Bernstein & Vazirani and Aaronson as its special cases. The RFS is a computational problem which shows the advantage of a quantum algorithm over its classical counterpart. The manuscript provides a new description of quantum algorithms in phase space, using the conjugate pair oracles.

The topics of quantum (un)computation and quantum algorithms are current and consistent with the research interest of the journal. I recommend the manuscript for publication but only after the authors address consider some points.

1) The author should clearly mention some possible applications of the results. Is it possible to apply to quantum search algorithms? Quantum walks?

Somehow related to this point: although the topic is current there are only two references from the last five years (both of them from the authors).

2) The Boolean function f is chosen to be linear (Eq. (2)). The authors should explain better this particular choice and mention how (if) the conclusions will change if other type of functions are chosen.

3) It is not too clear the difference between U_{f_k} and U_{g_k}. The only place when the later is mentioned is on page 6, line 149, without too many details.

4) Overall the manuscript is well written. However, in the first paragraph of page 2, line 41, the second “bounds” seems wrong.

Round 2

Reviewer 1 Report

Comments and Suggestions for Authors

No more comments.